# “Slash and Clear”, a Community-Based Vector Control Method to Reduce Onchocerciasis Transmission by *Simulium sirbanum* in Maridi, South Sudan: A Prospective Study

**DOI:** 10.3390/pathogens10101329

**Published:** 2021-10-15

**Authors:** Stephen Raimon, Tom L. Lakwo, Wilson John Sebit, Joseph Nelson Siewe Fodjo, Peter Alinda, Jane Y. Carter, Rory J. Post, Robert Colebunders

**Affiliations:** 1Amref Health Africa, Juba P.O. Box 410, South Sudan; Stephen.Jada@amref.org; 2Vector Control Division, Ministry of Health, Kampala P.O. Box 1661, Uganda; tlakwo@gmail.com (T.L.L.); papeteralinda@gmail.com (P.A.); 3Public Health Laboratory, Ministry of Health, May Rd, Juba P.O. Box 30125, South Sudan; wilson.sebit@hotmail.com; 4Global Health Institute, University of Antwerp, 2610 Antwerp, Belgium; JosephNelson.SieweFodjo@uantwerpen.be; 5Amref Health Africa Headquarters, Nairobi P.O. Box 27691-00506, Kenya; Jane.Carter@Amref.org; 6Disease Control Department, London School of Hygiene and Tropical Medicine, Keppel Street, London WC1E 7HT, UK; R.J.Post@ljmu.ac.uk; 7School of Biological and Environmental Sciences, Liverpool John Moores University, Liverpool L3 5UG, UK

**Keywords:** onchocerciasis, *Simulium damnosum*, *Simulium sirbanum*, vector control, slash and clear, elimination, community

## Abstract

Background: High ongoing *Onchocerca volvulus* transmission was recently documented in Maridi County, South Sudan. To complement community-directed treatment with ivermectin (CDTI) as the main onchocerciasis control strategy, we initiated a community-based vector control method “slash and clear” at the Maridi dam, a *Simulium damnosum* s.l. breeding site, to reduce *O. volvulus* transmission. Methods: *Simulium damnosum* s.l. biting rates were collected before and during the twenty months following the “slash and clear” intervention using the human landing catches. Black flies were dissected to measure parity rates before and twelve months after the intervention. Larvae and pupae of *S. damnosum* s.l. were collected from the dam for morphological and chromosomal characterization to identify the cytospecies involved. Results: Biting rates of *S. damnosum* s.l. close to the Maridi dam spillway decreased by >90% post-“slash and clear” for more than six months. Twelve months after the “slash and clear” intervention, the reduction in biting rates was still at <50% (*p* = 0.0007). Parity rates reduced from 13% pre-“slash and clear” (November 2019) to 5.6% post-“slash and clear” (November 2020). Larvae collected from the dam were identified as *Simulium sirbanum*. Conclusion: The “slash and clear” method was found to be an effective and cheap community-based method to reduce black fly biting rates caused by *S. sirbanum*. When repeated at least annually together with a high CDTI coverage, this intervention has the potential to considerably accelerate onchocerciasis elimination.

## 1. Introduction

Onchocerciasis is a disease of major public health importance in Maridi County, South Sudan, where it is associated with a high prevalence of epilepsy, including nodding syndrome [1,2]. Half the adult volunteers tested for onchocerciasis in the Maridi villages were found to have *Onchocerca volvulus* microfilariae in their skin snips, while over 80% of persons with epilepsy (PWE) were also infected [3]. Communities living close to the Maridi dam more often complained of intense black fly nuisance and were more affected by seizure disorders. In 2018, in a door-to-door survey, an epilepsy prevalence of 4.4% and annual incidence of 373.9/100,000 person years were documented in villages close to the Maridi River, with the highest prevalence (11.9%) observed in Kazana 2, a village close to the Maridi dam [1]. For many years, mass drug administration of ivermectin had been interrupted in the area and only 40.8% of the population took ivermectin in 2017 [1].

Vector control using larvicides was the primary strategy of successful onchocerciasis control programs in West Africa which operated from 1974 to 2002 [4]. However, the implementation of larvicide treatments is labor-intensive, costly and potentially harmful to the environment. Moreover, insecticide resistance is a concern for the long-term success of this method. A study in multiple communities in northern Uganda using a “slash and clear” method resulted in an 89−99% decline in vector biting rates that lasted up to 120 days post-intervention [5]. We investigated the feasibility and efficacy of this “slash and clear” method in reducing black fly biting rates at the Maridi dam, the only black fly breeding site identified in the Maridi central area.

*Simulium damnosum* s.l. is the only known vector of onchocerciasis in South Sudan and occurs in most rapidly flowing rivers and streams [6], including the Maridi River (or River Gel) at the Maridi dam [7,8]. The precise cytospecies within the *S. damnosum* complex present at Maridi has never been described, although both *S. damnosum* s. str. and *S. sirbanum* have been identified from tributaries of the Bahr el Ghazal River, northwest of Maridi [8], and from rivers in northern Uganda (Post, unpublished data).

## 2. Materials and Methods

### 2.1. Study Area

Maridi County is situated in Western Equatoria State of South Sudan, and its main river is the Maridi River (known as the River Gel further downstream). The study area was extensively described in a previous publication [9]. Community-directed treatment with ivermectin (CDTI) coverage in the study villages was estimated at 75.7% in 2019 [10]. An entomological assessment carried out in December 2019 found that the Maridi dam was the only black fly breeding site in the area, with several larvae and pupae found on vegetation on the dam spillway (Figure 1, Panel A), and this confirms previous reports that there were no biting *Simulium damnosum* s.l. in the area before the dam was built in 1958 [7]. Moreover, only three communities along the Maridi River experienced intense black fly biting compared to the rest of the population [9]. Vector breeding had been subject to previous successful control by periodically pouring spent engine oil over the dam spillway [8], but this must have been environmentally damaging and seems to have been discontinued sometime after 1985.

### 2.2. Study Procedures

#### 2.2.1. Identification of the *O. volvulus* Vector in Maridi

Larvae and pupae were collected from the riverine vegetation at the Maridi dam on 07 December 2020 and preserved in Carnoy’s fixative according to standard procedures [11]. Specimens were identified morphologically as *S. damnosum* s.l. according to Crosskey [12], and stored at 4 °C. Final instar larvae were stained using both feulgen (Schiff’s reagent) and lacto-propionic-orcein in the laboratory according to standard procedures [11]. Polytene chromosomes were examined and identified with reference to their inversions [13,14]. Two preserved, mature male pupae (which are black shortly before hatching) were dissected to extract the adult male fly, which was allowed to surface-dry, and the scutal pattern was characterized according to the standard classification [15].

#### 2.2.2. Human Landing Catches (HLC) of Adult Biting *Simulium* along the Maridi River

Three black fly catching sites were established for this study: at the Maridi dam (20 m from the spillway), Kazana 2 (600 m from the dam), and Matara (3.5 km downstream from the dam on the Maridi River). In each catching site, two vector collectors worked on alternate hours according to HLC standard procedures [6,7].

*Simulium* fly catches were conducted for seven consecutive days before the “slash and clear” intervention to establish a baseline, and on a weekly basis for 20 months after the intervention. Trained catchers sat on a chair or log with their legs bare below the knees and any flies landing on their legs were caught by inverting a small plastic tube over them. Catches were conducted between 7:00 and 18:00 h, and the number of flies caught on an hourly basis was recorded in a pre-designed form. The fly biting rate was then calculated.

#### 2.2.3. Dissection of Flies for Age Group Determination (Parity)

Some adult biting *S. damnosum* caught by HLC were kept alive by wrapping them in a moist towel wetted with water. Flies were transported to the Maridi Health Sciences Institute where a room was assigned as a temporary laboratory. Flies were killed by ether vapor, which substituted for chloroform [16], before placing them on a slide which was transferred to a stereo-microscope. The species was confirmed before opening the abdomen. Dissection for parity was performed according to standard routine procedures [17,18] and flies were categorized as nulliparous or parous.

#### 2.2.4. “Slash and Clear” Activity

Local volunteers aged 18–30 years residing close to the Maridi dam were identified by the local authorities and recruited to participate in the “slash and clear” intervention. The volunteers were initially taken through theoretical training on black flies, how to locate breeding sites, types of breeding sites, and how to slash trailing vegetation using machetes. The trainees were then brought to the breeding site at the Maridi dam and instructed on the process of scraping the trailing vegetation off the dam spillway using locally made tools and throwing the vegetation onto the river bank to dry, thereby killing the adherent fly larvae and pupae (Figure 1, Panel A). The techniques and procedures were adopted from those described by Jacob et al. [5]. Other vegetative growth on surrounding rock surfaces at the dam spillway were also cleared to destroy all possible breeding sites. The entire process was accomplished in three days (6, 7, and 9 December 2019), with eight volunteers working for a minimum of six hours a day. These volunteers from the Maridi community were supervisied by an experienced entomologist skilled in the “slash and clear” technique. A second round of the “slash and clear” intervention was implemented in December 2020 (18, 19, 20, and 21 December 2020), supervised by the South Sudannese national entomologist (W.J.S).

### 2.3. Data Analysis

Monthly biting rates (MBR) were calculated as number of flies collected during the month divided by the number of catching days in the month and multiplied by the number of days in the month, as previously described [12]. The parous rate was defined as the percentage of parous females in a sample of dissected female flies in a defined site. Data were entered into spreadsheets and analyzed using R version 3.5.1. Continuous variables were summarized as means and compared using the Mann–Whitney U test, while categorical variables were expressed as percentages and compared using Chi-squared tests. *p*-values below 0.05 were considered statistically significant.

## 3. Results

### 3.1. Identification of Vector Cytospecies Found Breeding at the Maridi Dam

For unknown reasons, larvae from the Maridi dam did not stain well for chromosomal examination, although other species, used as controls and processed simultaneously, stained very well. In view of this, only a small number of specimens of *S. damnosum* s.l. larvae from the Maridi dam could be identified, none of them could be fully karyotyped, and it was not possible to determine the sex of most larvae. However, ten specimens were identified and all of them were *S. sirbanum* because they were all homozygous for inversions 2L-C.8 [13,14]. Of these specimens, three could be scored for chromosome arm 1S, and were found to be homozygous for inversion 1S-3, and no heterozygotes were observed, including the only specimen which was stained sufficiently to be identified as male. The scutal pattern of both males dissected from pupae was ‘Type IV’.

### 3.2. Human Landing Catches of Adult Biting Simulium around the Maridi Dam

At baseline (November 2019), the monthly biting rates at the Maridi dam, Kazana 2, and Matara sites were 6038.6, 4697.2, and 3660, respectively (*p* = 0.024) (Table 1). These monthly biting rates decreased drastically by 99% immediately after the “slash and clear” intervention. However, in the second month post-intervention, there was a spike in MBR observed in all three sites (Maridi dam, Kazana 2, and Matara: 4347.9, 3278.3, and 2433.5, respectively). From February 2020, a gradual reduction in MBR was observed in all sites, which started rising again progressively from the seventh month post-intervention. Black fly-free months were from March to June 2020, where zero flies were registered at all three sites. Although a rise in MBR was observed from the seventh month, it was not comparable to baseline figures. Immediately after the second slashing in December 2020, the MBR decreased by more than 97%; seven months after the second slashing, the MBR remained 70% less than the December 2020 baseline (Figure 2).

### 3.3. Dissection of Flies for Age Group Determination (Parity)

Of the 408 adult female *S. damnosum* captured, 400 were dissected (Table 2). Of these, 13.0% were parous, with parity rate ranging from 13.1% to 15.7%. Nearly all the parous flies appeared to be young, with opaque malpighian tubules and fat bodies present. In a few cases, fresh blood was observed, an indication that they most likely had fed on a human being prior to capture (there are no cattle in the area, but some goats and sheep). No flies with retained eggs were observed. In the post-slash, out of the 446 flies dissected, 5.6% were parous. A reduction in parity rate was therefore generally observed.

### 3.4. Impact of the “Slash and Clear” Intervention

The “slash and clear” intervention resulted in a dramatic reduction in fly biting rates by >90% in most of the months post-intervention (January–July 2020). The impact was dramatic in the first month, although there was a spike in biting rate in the second month (January 2020). From the fourth month post-“slash and clear”, all the sites registered zero flies, which persisted for a period of four months. Slow vector repopulation only started to be seen from the seventh month post-intervention (Figure 2); however, the population recovery was only 50% of that at baseline. After the second slashing in December 2020, the biting rate reduced by more than 97% from the December 2020 level, and seven months later, the MBR remained 70% below the December 2020 level and 88.1% below the November 2019 baseline. Twelve months after the first “slash and clear” intervention, and seven months after the second “slash and clear”, there was slow re-growth of vegetation observed at the Maridi dam spillway (Figure 1, Panel C).

### 3.5. Cost of the “Slash and Clear” Intervention and Entomological Assesment

The incentive for the slashers was $360 (United States dollars) ($10 a day to 9 volunteers for 4 days), and for the fly-catchers for the seven days was $490 ($10 a day to 6 catchers and 1 supervisor for 7 days of catching), and another $1500 for purchase of tools and supplies (machetes, gumboots, locally made scraping tools, rain coats, catching tubes, hand lenses, etc.).

## 4. Discussion

The chromosomal characterization of the larvae collected from the Maridi dam clearly indicates that *S. sirbanum* is the main (or only) vector species in the area. Furthermore, chromosomal characterization is consistent with the resident cytotype being the ‘Type IV form’ of *S. sirbanum* [19]. However, the small number of specimens examined cannot rule out the possibility of the ‘Sudanese form’ of *S. sirbanum*, but this is unlikely because the two males extracted from pupae were found to have ‘Type IV’ scutal patterns, which is characteristic of the ‘Type IV form’, whilst the ‘Sudanese form’ usually has type III scutal patterns [20].

Our study shows that black fly biting rates in Maridi, South Sudan, were drastically reduced by >90% in all sites following implementation of the “slash and clear” exercise, and biting rates remained low for several months afterwards. This finding is very much in agreement with that of Baker and Abdelnur [20], who cleared vegetation from a single site on the River Bussere about 25 km upstream of Wau, where *S. sirbanum* represented around 90% of all larvae identified [8]. They did not measure the effect on biting rates, but they reported that breeding was “virtually eliminated”. More recently, in northern Uganda, Jacob et al. [5] reported that removal of trailing vegetation from *S. damnosum s. str.* breeding sites resulted in a dramatic reduction in biting rates (ca. 90%). However, for the Maridi dam, the trend in biting rates was slightly different from Uganda because of a spike observed during the first month post-intervention. This was likely due to some remaining mature larvae and pupae that might have been missed from being scraped from the concrete dam spillway, and these later hatched and contributed to the population of flies recorded in February 2020. This has provided a unique lesson when undertaking this intervention in ecological areas where *S. damnosum* s.l. is also breeding on concrete and rocks, whereas in rivers, where breeding is entirely on trailing vegetation, once this is removed, no residual black fly population is observed. However, the slow population recovery in *S. damnosum* s.l. observed seven months post-intervention was most likely attributed to the slow re-growth of vegetation on the spillway at the Maridi dam (Figure 1, Panel C). Vegetation provides support for attachment of young stages of the vector and directly impacts on the population at breeding sites, as was earlier reported also in Uganda [5]. The reduction in parity rate observed post-“slash and clear” intervention could be attributed to the success of this intervention, as the flies coming out are those newly hatched from the breeding sites. Lewis earlier reported that the percentage of nulliparous flies is expected to rise after heavy rains and when rivers have risen, and this is due to the emergence of fresh flies [17]. In the case of the sites at the Maridi dam, dissection for parity was performed towards the end of the rainy season, and the variation in parity rate pre- and post-intervention may require a prolonged study.

In Maridi, the “slash and clear” technique was rapidly accepted by the local population. Community volunteers were trained during the initial “slash and clear” intervention and were able to repeat the intervention satisfactorily with limited supervision the second time. Given the past nuisance caused by black flies in these villages and the high incidence of onchocerciasis-associated epilepsy, the community is highly motivated to continue this intervention in the coming years. To further decrease black fly biting rates, the plan is to repeat the “slash and clear” intervention in the future every seven months.

A limitation of our study is that black fly infectivity rates could not be determined. Black flies were collected to be pool-screened with *O. volvulus* PCR, but because they had not been adequately preserved, this screening could not be performed.

Vector control is currently underutilized in onchocerciasis control programs in Africa due to inadequate resources for costly larviciding and lack of adequate entomological workforces [21], thus the need for research into other simpler vector control approaches. The recent innovation of “slash and clear” has provided an additional tool for control and elimination of onchocerciasis, and is advocated as an alternative strategy [13]. A recent modeling study showed that supplementing annual drug treatments with “slash and clear”, even if vegetation is cleared only once per year, can significantly accelerate the achievement of onchocerciasis elimination [14]. However, the feasibility and efficacy of the “slash and clear” method may depend on local vegetation and other riverine features. The Maridi dam was an excellent site for implementing this method because it was the only productive site of *S. damnosum* s.l. breeding, and in addition, is the main source of drinking water for Maridi town and the villages close to it, therefore raising concerns from communities about the application of larvicides. This environmentally friendly intervention is simple and inexpensive to perform, requiring equipment commonly found in rural communities in Africa [4]. However, further field studies should be carried out in other geographical settings in the African region with other vector cytospecies. A cluster randomized trial to determine the effectiveness of the “slash and clear” intervention on *O. volvulus* transmission is currently ongoing in Cameroon [22], along with further work in northern Uganda [23].

The high baseline biting rates observed at the Maridi dam were not surprising; indeed, dam spillways have been reported to be excellent black fly breeding sites in the past [24,25,26,27,28]. Such areas need to be prioritized for onchocerciasis elimination efforts, since ivermectin alone may not be sufficient to achieve elimination in the face of such productive breeding sites, and additional tools may be needed. A combination of both CDTI and community-directed removal of black fly larval attachment sites is therefore recommended in these settings.

A weakness of the study is that the dam spillway was the only black fly breeding site in the area. Therefore, it was not possible to include a control site that was untreated to adjust for overall variations in black fly numbers caused by climatic changes. Dam spillways are an artificial environment where the degree of breeding is closely associated with the amount of water flowing over the dam. However, no data about water flow were available at the Maridi dam. Since the insecurity in the area in 2013, the qualified staff at the dam have left. Since then, there have been no regular measurements of water levels at the dam reservoir, and the water gates of the dam were poorly managed to release water when there is water overspill. Therefore, the overspill of water through the dam spillway is continuous throughout the year, with higher levels in the rainy season (March to November) and minimal levels during the dry season (December to February). However, without the “slash and clear”, the vegetation growth is present throughout the year.

## 5. Conclusions

The “slash and clear” method was found to be a very effective, simple, and cheap community-based method to reduce biting rates caused by *S. sirbanum* from a local breeding site. When repeated at least annually together with high coverage of community-directed treatment with ivermectin, it has the potential to considerably accelerate onchocerciasis elimination.

## Figures and Tables

**Figure 1 pathogens-10-01329-f001:**
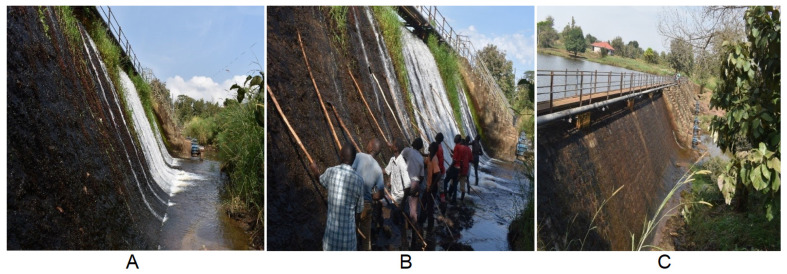
(**A**) Before “slash and clear” in November 2019, (**B**) during “slash and clear” at the Maridi dam spillway, and (**C**) post-“slash and clear” at the Maridi dam, December 2019.

**Figure 2 pathogens-10-01329-f002:**
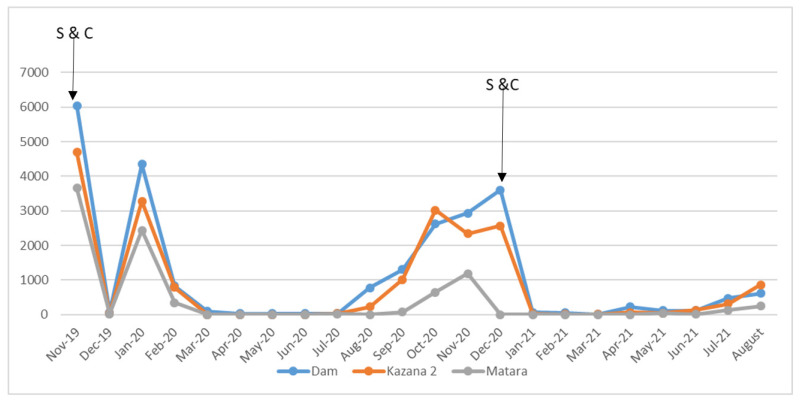
Graphical illustration of the evolution of monthly biting rates in Maridi, South Sudan.

**Table 1 pathogens-10-01329-t001:** Monthly biting rates (MBR) for the three catching sites in Maridi, South Sudan, pre- and post-“slash and clear” intervention.

Month	Maridi Dam		Kazana 2		Matara	
	Monthly Biting Rates	% Δ	Monthly Biting Rates	% Δ	Monthly Biting Rates	% Δ
November 2019 (baseline)	6038.6	NA	4697.2	NA	3660.0	NA
December 2019	62.0	99.0	31.0	99.4	23.4	99.4
January 2020	4347.8	28.0	3278.3	23.5	2433.5	33.5
February 2020	826.5	86.3	790.3	83.2	348	90.5
March 2020	93.0	98.5	0.0	100.0	0.0	100.0
April 2020	37.5	99.4	0.0	100.0	0.0	100.0
May 2020	31.0	99.5	0.0	100.0	0.0	100.0
June 2020	30.0	99.5	0.0	100.0	0.0	100.0
July 2020	31.0	99.5	31.0	99.3	23.3	99.4
August 2020	775.0	87.2	224.8	95.2	0.0	100.0
September 2020	1305.0	78.4	1020.0	78.3	75.0	98.0
October 2020	2627.3	56.5	3022.5	35.7	651	82.2
November 2020	2932.5	51.5	2340	50.2	1192.5	67.4
December 2020 (baseline)	1036.3	82.8	779.4	83.4	234.7	93.6
January 2021	62	94	23.25	99.5	0.0	100.0
February 2021	56	94.5	14	98.6	0	100.0
March 2021	7.75	99.2	7.75	99.2	0	100.0
April 2021	222	78.5	66	93.6	0	100.0
May 2021	124	88	46.5	95.5	38.75	96.2
June 2021	120	88.4	127.5	87.6	7.5	99.2
July 2021	477.4	54	316.2	69	130.2	87.4
August 2021	620	40.1	868	16.2	248	76

% Δ: Percentage change compared to baseline biting rate. NA: Not applicable.

**Table 2 pathogens-10-01329-t002:** Parity and infectivity rates of *S. damnosum* s.l. pre- and post-“slash and clear” intervention at the Maridi dam, South Sudan.

Catching Site	No. of Flies Caught	No. Dissected	No. Parous	Parity Rate (%)
Pre-“slash and clear”				
Maridi dam	143	140	19	13.1
Kazana 2	147	145	15	10.7
Matara	118	115	18	15.7
Total	408	400	52	13.0
Post-“slash and clear”				
Maridi dam	234	227	14	6.1
Kazana 2	176	169	10	6.0
Matara	53	50	1	2.0
Total	472	446	25	5.6

## Data Availability

The datasets generated and/or analyzed during the current study are available from the corresponding author upon reasonable request.

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
