# Peer review of "“Slash and Clear”, a Community-Based Vector Control Method to Reduce Onchocerciasis Transmission by Simulium sirbanum in Maridi, South Sudan: A Prospective Study"

_pathogens, 2021, doi:10.3390/pathogens10101329_

Round 1

Reviewer 1 Report

This manuscript reports the result of a trial of a community-based vector control measure (habitat removal of black fly larval and pupal attachment vegetation) at a dam spillway in South Sudan.  The results are similar to those seen in previous trials in Uganda and elsewhere, which report a dramatic reduction in black fly biting in nearby communities, with the effect lasting up to a year.  Importantly, this study was carried out on a dam spillway, while the prior studies were carried out in free flowing rivers.  This is significant, as dam spillways are recognized as prolificate breeding sites for the black fly vectors of Onchocerca volvulus.  The manuscript is clearly written and the results they report are carefully analyzed.  I have a few comments that I would like to see the authors address:

  1. One weakness in the study is that the dam spillway that was the subject of the study was the sole major breeding site in the area.Thus, it was not possible for the authors to include a control site that was untreated to adjust for overall variations in black fly numbers caused by climatic changes.  However, dam spillways are an artificial environment where the degree of breeding is closely associated with the amount of water flow over the dam.  Do the authors to any data on water flow at the dam that might be used for a surrogate for breeding site productive potential?

  1. The authors quote a cost for the slash and clear in the discussion, but it is not clear what the currency used is.$1500 USD to equip 8 volunteers with machetes and rubber boots seems rather high to me.

  1. The references do no correspond to the numbers in the text.The authors need to carefully review the reference pointers and reference list to ensure they match.

  1. In line 54, blackfly should not be italicized.

Author Response

Response to reviewer 1

Reviewer

This manuscript reports the result of a trial of a community-based vector control measure (habitat removal of black fly larval and pupal attachment vegetation) at a dam spillway in South Sudan.  The results are similar to those seen in previous trials in Uganda and elsewhere, which report a dramatic reduction in black fly biting in nearby communities, with the effect lasting up to a year.  Importantly, this study was carried out on a dam spillway, while the prior studies were carried out in free flowing rivers.  This is significant, as dam spillways are recognized as prolificate breeding sites for the black fly vectors of Onchocerca volvulus.  The manuscript is clearly written and the results they report are carefully analyzed.  I have a few comments that I would like to see the authors address:

  1. One weakness in the study is that the dam spillway that was the subject of the study was the sole major breeding site in the area. Thus, it was not possible for the authors to include a control site that was untreated to adjust for overall variations in black fly numbers caused by climatic changes.  However, dam spillways are an artificial environment where the degree of breeding is closely associated with the amount of water flow over the dam.  Do the authors to any data on water flow at the dam that might be used for a surrogate for breeding site productive potential?

Response 1

We now added as a weakness of the study

“A weakness of the study is that the dam spillway was the only blackfly breeding site in the area. Therefore, it was not possible to include a control site that was untreated to adjust for overall variations in blackfly numbers caused by climatic changes. Dam spillways are an artificial environment where the degree of breeding is closely associated with the amount of water flow over the dam. However no data about water flow was available at the Maridi dam. Since the insecurity in the area in 2013, the qualified staff at the dam left the area. Since then there have been no regular measurements of water levels at the dam reservoir, and the water gates of the dam were poorly managed to release water when there is water overspill. Therefore, the overspill of water through the dam spillway is continuous throughout the year with higher levels in the rainy season (March to November) and minimal levels during the dry season (December to February). However, without the “slash and clear” the vegetation growth is present throughout the year.”

 Reviewer

  1. The authors quote a cost for the slash and clear in the discussion, but it is not clear what the currency used is. $1500 USD to equip 8 volunteers with machetes and rubber boots seems rather high to me.

Response 2

The cost were calculated using United States Dollars, and the 1500 USD cost for equipment was for all supplies required to complete the entomological assessment of the river Maridi and to perform slash and clear

We now adapted the text in the paper as follows:

“3.5. Cost of the “slash and clear” intervention and entomological assesment. 

The field allowance for the slashers was 360 $ (United States dollars)(10 $ a day to 9 volunteers for 4 days) and for the fly-catchers for the seven days was 490 $ (10 $ to 6 catchers and 1 supervisor for 7 days catching), and another 1500 $ for purchase of tools and supplies (machetes, gumboots, locally-made scraping tools, rain coats, catching tubes, hand lenses, etc).”

 Reviewer

  1. The references do no correspond to the numbers in the text. The authors need to carefully review the reference pointers and reference list to ensure they match.

Response 3

We reviewed and corrected all the reference numbers

 Reviewer

  1. In line 54, blackfly should not be italicized

Response 4

We corrected this mistake

Reviewer 2 Report

I read this short manuscript with great interest.  The manuscript essentially reports a low technology community based solution to controlling black flies.  The methods and results appear to yield measurable results that might have real world implications for the control of onchocerciasis.

I think the report itself is a well written summary of observable data on a limited number of community based control programs in a small area.  The major limitation is a lack of mirrored sites to compare the treatment with.  This leads to the ever present problem in a situation like that presented here where the vector population changes might be due to other local or area wide environmental issues that have nothing to do with the community led vector control.  I realize and I am sure the authors realize that paired programs with similar habitats are often difficult or logistically impossible to run.  It is a limitation , however.

I think the other major limitation was alluded to in line 195 and then again in line 238.  This is the problem where the vectors of the filarial nematodes are not actually verified.  In addition, there is no verification that the rate of infectious biting vectors was reduced only that the flies presumably rising off the dam site are being controlled.  If by an unlikely chance the vector of onchocerciasis in this area is not the species from this dam then the control is only for nuisance purposes.  This limitation on the study was addressed and pointed out was due to inadequate specimen preservation.  I don’t think this prevents publication but it is a weakness due to the failure to show the rate of infected and possibly infectious flies were going down as control occurred.

I have a few minor comments.  I think "blackfly" and "blackflies" are better put as "black fly" and "black flies"

Also the "volunteers" e.g. line 191 are paid a salary so they are not really volunteers and are instead staff or something.

Author Response

Response to reviewer 2

Reviewer

I read this short manuscript with great interest.  The manuscript essentially reports a low technology community based solution to controlling black flies.  The methods and results appear to yield measurable results that might have real world implications for the control of onchocerciasis.

 I think the report itself is a well written summary of observable data on a limited number of community based control programs in a small area.  The major limitation is a lack of mirrored sites to compare the treatment with.  This leads to the ever present problem in a situation like that presented here where the vector population changes might be due to other local or area wide environmental issues that have nothing to do with the community led vector control.  I realize and I am sure the authors realize that paired programs with similar habitats are often difficult or logistically impossible to run.  It is a limitation , however.

Response

We now added in the discussion

“A weakness of the study is that the dam spillway was the only blackfly breeding site in the area. Therefore, it was not possible to include a control site that was untreated to adjust for overall variations in blackfly numbers caused by climatic changes. Dam spillways are an artificial environment where the degree of breeding is closely associated with the amount of water flow over the dam. However no data about water flow was available at the Maridi dam. Since the insecurity in the area in 2013, the qualified staff at the dam left the area. Since then there have been no regular measurements of water levels at the dam reservoir, and the water gates of the dam were poorly managed to release water when there is water overspill. Therefore, the overspill of water through the dam spillway is continuous throughout the year with higher levels in the rainy season (March to November) and minimal levels during the dry season (December to February). However, without the “slash and clear” the vegetation growth is present throughout the year.”

Reviewer

I think the other major limitation was alluded to in line 195 and then again in line 238.  This is the problem where the vectors of the filarial nematodes are not actually verified.  In addition, there is no verification that the rate of infectious biting vectors was reduced only that the flies presumably rising off the dam site are being controlled.  If by an unlikely chance the vector of onchocerciasis in this area is not the species from this dam then the control is only for nuisance purposes.  This limitation on the study was addressed and pointed out was due to inadequate specimen preservation.  I don’t think this prevents publication but it is a weakness due to the failure to show the rate of infected and possibly infectious flies were going down as control occurred.

Response

We agree that this is a limitation of the study and a new collection of black flies is planned in Maridi.

I have a few minor comments.  I think "blackfly" and "blackflies" are better put as "black fly" and "black flies"

Response

We now use black fly and black flies

Also the "volunteers" e.g. line 191 are paid a salary so they are not really volunteers and are instead staff or something.

Response

We now mentioned they were paid an incentive
